# Patient Reported Outcome Measures (PROMs) in Surgery: Evaluation after Minimally Invasive Reduction and Percutaneous K-Wires Fixation for Intra-Articular Calcaneal Fractures

**DOI:** 10.3390/diseases11020057

**Published:** 2023-04-05

**Authors:** Lorenzo Brognara, Antonio Mazzotti, Alberto Arceri, Elena Artioli, Giacomo Casadei, Simone Bonelli, Francesco Traina, Cesare Faldini

**Affiliations:** 1Department of Biomedical and Neuromotor Sciences (DIBINEM), Alma Mater Studiorum University of Bologna, 40126 Bologna, Italyfrancesco.traina@ior.it (F.T.);; 21st Orthopaedics and Traumatologic Clinic, IRCCS Istituto Ortopedico Rizzoli, 40136 Bologna, Italy; 3Ortopedia-Traumatologia e Chirurgia Protesica e dei Reimpianti d’anca e di Ginocchio, IRCCS Istituto, Ortopedico Rizzoli, 40125 Bologna, Italy

**Keywords:** calcaneal fractures, minimally invasive, patient reported outcome measures (PROMs)

## Abstract

Background: The optimal surgical treatment of intra-articular calcaneal fractures (IACF) is still under debate. In the literature, results are based on clinical or radiographical findings. Few studies have evaluated the effect of patient expectations on patient-reported outcomes after surgery and little is known about outcomes directly reported by the patient who experienced it. Patient reported outcome measures (PROMs) may represent a viable and useful tool for evaluating the efficacy of the procedure and can be considered as an indicators of health-care quality. The aim of this study is to evaluate PROMs after minimally invasive reduction and percutaneous Kirschner-wires fixation for IACF, and to compare PROMs to pre-operative and last follow-up radiographic findings. Methods: 33 consecutive patients with IACF treated with minimally invasive reduction and percutaneous K-wires fixation were included. Data collection included demographics, pre-operative and last available Böhler and Gissane angle X-rays, foot function index (FFI), and foot and ankle outcome score (FAOS). Results: At a mean follow up of 36.7 months, the mean FFI score was 24.3 ± 19.9 and the mean FAOS score was 68 ± 24.8. Patients with better Gissane angle showed better activity limitations FFI subscores. Moreover, worse pre-operative Gissane and Böhler angle were significantly associated with a worse total FAOS score and subscores. Conclusions: Minimally invasive reduction and percutaneous K-wires fixation provided satisfactory PROMs. Despite these results, prospective randomized studies are required to confirm the validity and reliability of PROMs in evaluating different treatments.

## 1. Introduction

Calcaneal fractures (CF) are the most common tarsal bone injury. They are intra articular in more than 70% of cases, involving the subtalar joint [1]. The optimal surgical treatment of IACF is still under debate, since many lifelong debilitating sequelae may occur [2,3,4]. CF frequently occur in young patients, who may be unable to return to their previous activities, therefore representing a relevant social and economic concern [2,3,4]. In spite of the high incidence of these injuries, their management is still debated [2,3]. Many classifications have been proposed for intra articular CF (IACF). Essex-Lopresti divided IACF into two categories: joint depression type and tongue type. These two configurations share a common primary fracture line and are distinguished based on a secondary fracture line—that is, when a tongue-type fracture occurs, the secondary fracture line longitudinally promulgates and exits posteriorly from the calcaneal tuberosity, below the level of the Achilles tendon insertion. Another well-known classification for IACF is the one proposed by Sanders, which is based on the number of intra-articular fracture lines and their location on semi-coronal CT images. This four-type classification is useful in understanding typical fracture patterns of the calcaneus and in predicting the outcomes. Type 4 classifications are severely comminuted, involving 3 or more fracture lines with greater than 2 mm of articular displacement, and are commonly associated with unfavorable results in terms of clinical outcomes.

When evaluating a IACF on standard lateral-view X-rays, the Böhler and Gissane angles are the main parameters to be considered, as specified in previous studies [3,4,5]. The Böhler angle is measured on a lateral foot radiograph between a line joining the highest point of the anterior process of the calcaneus and the highest point of the posterior articular facet, and a line joining the highest point of the posterior articular facet with the highest point of the calcaneal tuberosity. The Gissane angle is measured by drawing lines along the superior surfaces of the anterior process and the posterior facet of the calcaneus to meet at the calcaneal sulcus.

Non-operative treatment for intra-articular IACF was favored in the past, but always related to unsatisfactory results over time with functional loss and disability. For this reason, conservative treatment is now rarely considered. Furthermore, a study [6] suggested that surgical treatment of IACFs was also economically advantageous compared with conservative management, indeed an estimate of the direct and indirect health care costs associated with these two types of treatment management reported a lower rate of subtalar arthrodesis associated with surgical treatment, and also a shorter duration of time off work. Sanders type III and IV (multi-fragmentary joint fracture) commonly end up to post-traumatic subtalar osteoarthritis requiring a subtalar fusion over time: if initially treated with open reduction and internal fixation (ORIF), these fractures seems to obtain better results and lower wound complication rates after a secondary subtalar fusion compared to patients conservatively managed at first. This could be due to the difficulty in restoring calcaneal height and the talocalcaneal relationship in patients with calcaneal malunion as result of initial nonoperative treatment [7].

As a matter of fact, surgery aims to restore the congruency of the subtalar joint—and width, height, profile and alignment of the calcaneal bone—as to avoid medial and lateral soft tissue impingement and allowing the patient to resume acceptable socio-professional activities [3].

Several surgical treatments have been proposed [1,2,3,4,5,6]: ORIF, reduction and fixation through mini-incisions, external fixation, calcaneoplasty and arthroscopically-assisted procedures. All of these methods appear to achieve satisfactory outcomes [3].

ORIF has been the most performed technique in the past, and still represents the standard of care in many settings.

The last 3 decades traditional approach for addressing IACFs through ORIF consists of an extensive L-shaped lateral approach to the calcaneus which allows an optimal fracture exposure, visualization of the posterior facet joint and calcaneus lateral wall. However, despite the good results reported in the literature, this approach is associated with high complication rates, such as wound infection and dehiscence, tissue de-vascularization, injury of the sural nerve, and implant-related pain. [1,2,5,6]. As a matter of fact, the development of major wound complications is a serious complication because the soft-tissue envelope around the calcaneus is particularly thin and vulnerable. Soft tissue impairments including wound edge necrosis, dehiscence, hematoma or deep infection remain frequent, hypothetically due to a large surgical field, long surgical time, dead space formation and eventual lateral calcaneal branch lesion of the peroneal artery, which vascularly supply the overlying fascio-cutaneous flap.

To overcome these disadvantages, percutaneous and minimally invasive approaches have been proposed, including external fixation, arthroscopically assisted procedures, calcanealplasty, as well as reduction and fixation through a small sinus tarsi approach (STA) [3,4,5,6,7,8].

Concerning the percutaneous approach, several indirect reduction techniques were proposed such as the Essex-Lopresti method using a percutaneous lever to manipulate and a Schanz pin introduced percutaneously to fix the main calcaneal fragment [7]. In addition, all variations of Essex-Lopresti method [9,10], a three-point distraction system to control the fragments separately [11], a combination of articular surfaces reduction through direct percutaneous method aided by a Steinmann pin and indirect reduction through traction and compression of the calcaneal posterior tuberosity [12] were also proposed. For the fixation, percutaneous cannulated or no-cannulated screws and percutaneous Kirschner-wires were suggested. All these percutaneous reduction and fixation methods reported generally good results.

External fixation minimize skin complications, restore a good calcaneal shape, and allow early weight-bearing [3].

Arthroscopically assisted procedure may shorten hospital stay, reduce wound complication risks, radiation exposure and even allow constant joint surface monitoring during the entire surgery, and possibly, the performing of a joint surface debridement; however, this requires a long learning curve, more surgical time and is more expensive [3].

Calcaneoplasty was used in achieving promising outcomes in terms of bone healing and complication rates; however, poor evidence studies exist [13,14].

The STA has been advocated as an alternative to the traditional lateral extensive approach and consists of a mini-incision about 2–4 cm over the sinus tarsi that does not violate the angiosome of the lateral hindfoot like the lateral extensive approach. In addition, the same incision may be used later if subtalar arthrodesis or tendon debridement must be performed. This approach is advantageous compared to extensive lateral approach in terms of a shorter surgical time, lower rate of soft tissue problem and wound complications, with comparable outcomes and no differences in the postoperative Böhler and Gissane angles reported [6].

Thus, it is imperative to perform proper surgery and achieve good patient outcomes in order to consider: the patient factors, the fracture pattern and the surgical timing. Furthermore, the surgeon experience is important as smaller incisions or percutaneous approaches do not allow a wide calcaneal exposition and easy manipulation of the fracture, especially for percutaneous techniques in which the reduction of the fracture must be performed indirectly.

Despite the theoretical benefits of modern operative techniques, today there is no clinical evidence to determine the superiority of minimally invasive surgical approaches over others [1,2,3,4,5,6]. 

Most of the studies in the literature regarding IACF show results based only on clinical or radiographical findings. However, considering that the best treatment for IACF remains controversial, patient reported outcome measures (PROMs) may represent a viable and useful tool for evaluating the procedure’s efficacy. 

PROMs are commonly used to assess a patient’s health status, and are increasingly recognized in clinical practice, providing physicians with patients’ perceptions and views of their health. Understanding the health state of the patient is crucial because it can directly affect their quality of life and their satisfaction with surgical procedures. 

Traditional outcome measures focus on specific clinical outcomes, or postoperative rates, but there is an increasing interest in the patient’s viewpoint in order to assess the effectiveness of intervention in terms of treatment of symptoms and improvement in function. 

Evaluating patient satisfaction is important on many levels. The achievement of patient satisfaction leads patients to better follow their doctors’ instructions regarding rehabilitation therapy as well as medications. In addition, one of the main goals of physicians should be to provide high-quality care that satisfies the needs of patients. 

Therefore, the aim of this study is to evaluate PROMs after minimally invasive reduction and percutaneous Kirschner-wires (K-wires) fixation for IACFs, comparing them with demographic and radiographic findings.

## 2. Materials and Methods

We retrospectively reviewed a series of consecutive displaced IACF treated by a single surgeon with minimally invasive reduction and percutaneous K-wires fixation from 2015 to 2018. The inclusion criteria consisted of patients with a closed and unilateral IACF, who had undergone open reduction and percutaneous K-wires fixation. Patients were excluded against the following exclusion criteria:Bilateral calcaneus fracture;Associated fractures;Conservative treatment;History of prior or subsequent lower limb surgeries;Severe comorbidities (uncontrolled diabetes, rheumatic diseases, vascular disruptions);Incomplete radiographic imaging assessment.

Radiographic evaluation was performed pre-operatively and postoperatively at the last follow-up through X-rays. PROMs were administered at the last follow-up. 

The study was conducted in compliance with the Health Insurance Portability and Accountability Act and the Declaration of Helsinki (Reference: n° 659/Sper/IOR).

### 2.1. Surgical Procedure

The patients were positioned in lateral decubitus on a radiolucent table. A proximal thigh tourniquet was applied during surgery. A 2–4 cm skin incision was centered on the sinus tarsi. The fracture site was debrided of the hematoma to enable better visualization of the subtalar joint. Both direct and indirect reduction methods were used to achieve reduction. Calcaneal height was restored by traction the main calcaneal tuberosity fragment, which was subsequently stabilized with a percutaneous K-wire. The procedure widened the subtalar joint space, which made reduction easier. Subtalar joint sinking was reduced by leveraging fracture fragments with a small periosteal elevator through the skin incision. Once the reduction was achieved and confirmed by fluoroscopy, the anterior process and posterior facet fracture fragments were fixed with K-wires (Figure 1).

Postoperatively, a non-weight-bearing plaster cast was applied and maintained for 40 days. After this period, the K-wires were removed and the plaster cast was replaced with a non-articulated walker boot. At 50 days after surgery, patients were allowed to remove the walker boot temporarily and undertake active and passive movements of the ankle. Weight-bearing using the walker boot was allowed 2 months after surgery, and free weight-bearing without the boot started gradually 80 days after surgery.

### 2.2. Data Collection

Demographics of the selected patients were extracted from our institutional digital database, as well as early and late complications. 

Imaging of the selected patients was recovered from the digital database. IACF was pre-operatively classified according to Essex Lo Presti classification on the X-Rays and according to Sanders classification on computed tomography (CT) imaging, when available [15,16,17,18,19]. 

Measurements were obtained by two orthopedic surgeons on pre-operative and last follow-up radiographs. Inter-observer reliability was assessed.

Clinical outcomes were assessed through the following PROMs: foot function index (FFI) [20] and foot and ankle outcome score (FAOS) [21] questionnaires.

The FFI is a 23-item PROM questionnaire, widely used by clinicians and investigators to quantify the impact of foot impairment on disability, pain and activity limitation, which is divided into 3 subscales: pain, disability and activity limitations useful. Items are rated on a visual analogue scale (VAS) with higher scores indicating worse outcomes. Scores are summed and converted to a 0 (worst) to 100 (best) scale.

The FAOS consists of 42 items across 5 categories: pain, symptoms, activities of daily living (ADL), sport and recreation, and foot- and ankle-related quality of life (QoL). Each item is scored with standardized answer options on a five-point Likert scale. Sum scores are converted to a normalized 0 (worst) to 100 (best) scale separately for each category.

All subjects participated voluntarily, and informed consent to participate and release personal data for publication was obtained from all individuals included in the study.

### 2.3. Statistical Analysis

The statistical unit was represented by the single operated calcaneus. Continuous variables were reported as mean ± standard deviation (SD) and range. Categorical variables were summarized as frequencies and percentages. The Shapiro-Wilk test was used to verify normal distribution of data. 

Patients’ PROMs were stratified as good or poor, considering a cutoff of 60 point (good if > 60). 

Differences were evaluated using two tailed Student *t*-tests, or Mann-Whitney U-tests for unpaired groups or chi-squared tests, when appropriate. *p* value inferior to 0.05 was considered statistically significant. Data collection was performed using Microsoft Excel (Microsoft Corporation, Redmond, WA, USA) for Windows 10. SPSS Statistics version 23 (IBM Corp., Armonk, NY, USA) was used for data analysis.

Inter-observer reliability for radiographic measurement was assessed using the intraclass correlation coefficient (ICC).

## 3. Results

The institutional digital database was searched for eligible patients and 33 patients were enrolled for the study. In total, 16 patients (48.5%) were females and 17 (51.5%) were males. The mean age at surgery was 48.9 ± 13.7 (25–78) years. Postoperative complications occurred in 4 (12%) patients: 1 delayed wound healing solved with standard wound care, and 3 secondary osteoarthritis. 

Average follow-up for radiographic findings was 8.4 ± 3.7 (4–26) months, while the mean follow-up for PROMs collection was 36.7 ± 8.8 (28–51) months.

Pre-operative X-rays showed a tongue type fracture pattern in 14 (42.4%) patients, while a joint depression type was reported in the remaining 19 (57.6%) cases (Figure 2).

Böhler and Gissane angles measured in pre-operative and last follow-up radiographs are reported in Table 1. Inter-observer reliability was verified for the Böhler and Gissane angle with good results (ICC superior to 0.9) in both preoperative and last follow-up X-rays.

The mean FFI score at follow-up was 24.3 ± 19.9 (0.4–80.9) and the mean FAOS score was 68 ± 24.8 (8.9–100). Results are detailed in Table 2 and Table 3.

Significant differences in activity limitations FFI subscores were observed in relation to Gissane angle, in both preoperative and last follow-up X-rays. In particular, patients with lower Gissane angle showed better activity limitations FFI subscores. Moreover, high pre-operative Gissane angle was significantly associated with a worse total FAOS score and worse pain, symptoms and ADL FAOS subscores. Similarly, a low pre-operative Bohler angle was significantly associated with worse symptoms, ADL, sport and recreation FAOS raw subscores. Results are detailed in Table 4 and Table 5. 

Patients with the lowest PROM scores presented with complications: one patient with delayed wound healing and one with secondary osteoarthritis. 

PROMs were not affected by main demographic parameters, in particular variables such as age at surgery and at the PROMs collection, follow-up period and sex did not determine significant differences in PROMs, both in subscores and total scores.

## 4. Discussion

Although significant improvements in surgical technique have been made in recent years, the optimal surgical treatment for IACF remains controversial [1,2,3,4,5,6]. Several approaches and fixation methods have been described for open surgery: medial, lateral, sinus tarsi, double-incision (medial and lateral) and extensile lateral approaches. The extensile lateral approach has gained popularity in the last few decades but with critical issues due to wound complications.

Minimally invasive techniques, such as STA reduction and percutaneous K-wires fixation, combine the advantages of subtalar joint direct visualization, which allows good anatomical reduction, with minimal hardware implantation and the reduction of wound complications [3]. 

The risk of complications and infection seems to be very low compared with open techniques. Despite these theoretical advantages, no evidence of clinical and radiographical superiority has been demonstrated yet. For this reason, further outcome evaluation methods such as PROMs may help in improving current services and treatments. 

Many self-reported questionnaires have been proposed in the literature, including the orthopedic trauma setting [9,22,23], but few still about PROMs after surgical treatment for IACF [24]. In particular, Siebe De Boer et al. investigated both functional and PROMs from the three different treatment strategies for displaced IACF. Overall, ORIF resulted in superior functional outcomes and greater patient satisfaction, with tolerable complication rates and no revision surgery required. Biz et al. found that patients treated with open reduction and internal fixation techniques presented overall superior radiographic and functional outcomes compared with percutaneous approach [25,26]. Several minimally invasive techniques were developed for management of IACF; however, Ahmed Shams et al. reported that treatment of displaced IACF with minimally invasive reduction and fixation using either cannulated screws or K-wires can achieve similar excellent functional and radiological outcomes, with high patient satisfaction, and additionally, K-wires has the advantage of reduced operative time [27].

In our study, the validity of PROMs was confirmed by the fact that both the used scores were not affected by the demographic parameters.

Although TC evaluation was performed in many cases, we have solely considered the angular parameters on X-Ray because, in our experience, they allow for the evaluation of results in relation to the calcaneus morphological reconstruction (congruency of the subtalar joint, and width, height, profile and alignment). Postoperative Gissane and Böhler angle values were comparable to those reported in previous studies [3,5,6,28].

Therefore, PROMs were compared with Gissane and Böhler angle values—and interestingly, better angular values were found to be statistically significant associated with better postoperative PROMs.

Similarly, Loucks and Buckley [29,30] found some correlations between Böhler angle and PROMs, using Short Form 36 (SF-36) [31] and visual analogue scale (VAS) [32]. On the contrary, a recent study by Nooijen et al. [33] showed no significant correlations between radiograph parameters and the American orthopedic foot and ankle score (AOFAS) [34] and FFI score. However, these patients were treated with open reduction and internal fixation through an extended lateral approach, which may have substantially affected the results. Likewise, Su et al. [35] reported the results of 328 displaced IACF treated with different surgical techniques, sometimes with plate and screws through a traditional L-shaped extended lateral approach, and sometimes through a minimally invasive approach using an anatomic plate and multiple compression bolts. Despite the postoperative Bohler angle having a significant positive role in predicting the functional recovery, a weak correlation between the postoperative Böhler angle and the AOFAS score was reported. 

A poor correlation between the Böhler angle and PROMs was observed in patients treated with more invasive techniques [33,34], probably because of the known complications of standard surgical techniques [2,6,9]. Whereas the results of this study suggest that minimal invasive reduction and percutaneous K-wires fixation through a STA could offer advantages in both radiological outcomes and PROMs. 

Some limitations must be addressed, which include: the retrospective design, the lack of comparative groups and the short follow-up periods, which may introduce statistical bias. Moreover, no floor or ceiling effects of the collected PROMs were affected. 

On the other hand, the fact that the procedure was performed by the same surgeon in a relatively short time-frame guarantees good homogeneity in the surgical technique. PROMs and radiological outcome assessment seems to be promising and can be used for future randomized controlled trials or prospective cohort studies that focus on the same conditions. Considering our results, we believe that minimally invasive reduction and percutaneous K-wires fixation approach for IACF expands the indications of surgery for calcaneal fractures, with fewer treatment-related complication directly reported by the patient who experienced it.

### Limitations and Future Direction

Studies with larger samples, which include in their methods different surgical procedures, are still needed to better investigate the benefits of IACF treated by minimally invasive reduction and percutaneous K-wires fixation in order to draw more reliable conclusions.

This study is not without limitations. One limitation of the present study concerns the evaluation of effects of surgical procedure on PROMs, as the long-term effects have been evaluated at 36 months (mean). For these reasons, further studies should evaluate the effects at one year in order to better understand the short-term impact of this surgical approach.

## 5. Conclusions

Minimally invasive reduction and percutaneous K-wires fixation for IACF provided satisfactory PROMs. A statistically significant correlation between lateral radiograph calcaneal morphology and PROMs was also detected.

The minimally invasive STA and K-wires fixation showed the advantages of the direct visualization of the subtalar joint that allows for a better anatomical reduction, and of the minimal hardware implantation, which reduces post-operative wounds complications and intolerance. The essential conditions in order to achieve satisfactory outcomes include proper knowledge of the fracture type and careful pre-operative planning.

PROMs are increasingly applied in clinical practice, revealing patients’ views on their health status and well-being. Adoption of PROMs in clinical practice, particularly after surgery, can help physicians make decisions for individual patients [36]. This possibility should be mostly considered when the pathological condition represents a relevant social and economic concern.

Despite these promising PROMs outcomes regarding STA reduction and percutaneous K-wires fixation for IACF, prospective randomized studies with larger cohorts and longer follow-ups are required to better evaluate both the drawbacks and the advantages of each technique and confirm the validity and reliability of PROMs in evaluating different treatments.

## Figures and Tables

**Figure 1 diseases-11-00057-f001:**
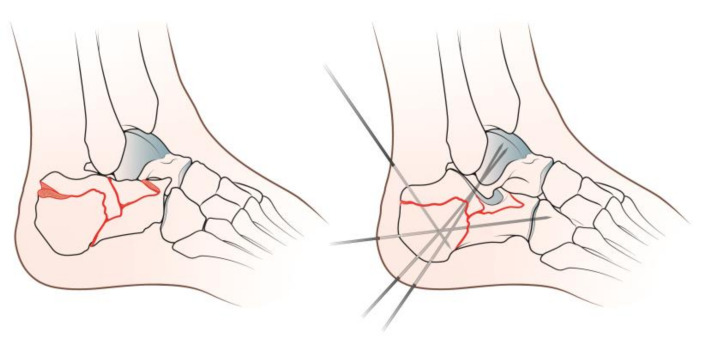
Graphical representation of the surgical procedure.

**Figure 2 diseases-11-00057-f002:**
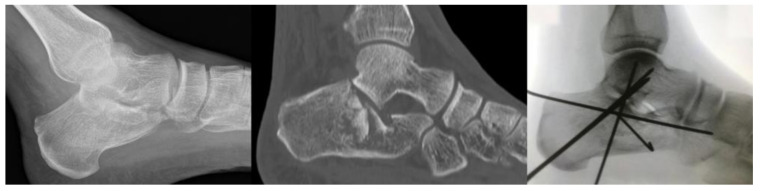
Left: pre-operative lateral X-Ray. Middle: pre-operative CT scan showing the IACF with severe joint depression. Right: postoperative lateral X-Ray showing good reduction stabilized with K-Wires.

**Table 1 diseases-11-00057-t001:** Böhler and Gissane angle measurements.

	Preoperative (Degrees) Mean ± SD (Range)	Last Radiographic F.U. (Degrees) Mean ± SD (Range)	*p*-Value
Böhler's angle ^+^	22.3 ± 15.1 (−15–55)	30.8 ± 12 (0–60)	<0.05
Gissane’s angle ^+^	126.3 ± 10.9 (110–150)	118.5 ± 9.4 (102–145)	<0.05

^+^ Student *t*-test.

**Table 2 diseases-11-00057-t002:** FFI (foot function index) score and subscores detailed.

	FFI			
	Pain (Raw Subscore)	Disability (Raw Subscore)	Activity Limitations (Raw Subscore)	Total (Normalized Score)
Score	24.7 ± 21 (0–78)	22.4 ± 19.1 (0–66)	6 ± 8.5 (0–44)	24.3 ± 19.9 (0.4–80.9)

**Table 3 diseases-11-00057-t003:** FAOS score and subscores detailed.

	FAOS					
	Pain (Raw Subscore)	Symptoms (Raw Subscore)	ADL (Raw Subscore)	Sport and Recreation (Raw Subscore)	QoL (Raw Subscore)	Total (Normalized Score)
Score	67.2 ± 30 (4–100)	66.2 ± 28.3 (3–100)	78.8 ± 23.2 (16–100)	44.7 ± 24.6 (5–100)	57 ± 30.9 (0–100)	68 ± 24.8 (8.9–100)

**Table 4 diseases-11-00057-t004:** The reported *p*-value with respect to different parameters, according to the patient’s stratification into good versus poor FFI normalized total score and subscores, considering their median value (median).

	FFI			
	Pain (Median = 27)	Disability (Median = 18)	Activity Limitations (Median = 4)	Total (Median = 20)
Age at surgery ***	NS	NS	NS	NS
Age at PROMs collection ***	NS	NS	NS	NS
PROMs F.U. ***	NS	NS	NS	NS
Sex °	NS	NS	NS	NS
Preoperative Gissane’s angle ***	NS	NS	*p* < 0.05	NS
Last F.U. Gissane’s angle ***	NS	NS	*p* < 0.05	NS
Preoperative Böhler's angle ***	NS	NS	NS	NS
Last F.U. Böhler's angle ***	NS	NS	NS	NS

* Mann-Whitney U-test, ° Chi-squared test.

**Table 5 diseases-11-00057-t005:** The reported *p*-value with respect to different parameters, according to the patient’s stratification into good versus poor FAOS normalized total scores and subscores, considering their median value (median).

	FAOS					
	Pain (Median = 79)	Symptoms (Median = 72)	ADL (Median = 87)	Sport and Recreation (Median = 45)	QoL (Median = 56)	Total (Median = 72)
Age at surgery ***	NS	NS	NS	NS	NS	NS
Age at PROMs collection ***	NS	NS	NS	NS	NS	NS
PROMs F.U. ***	NS	NS	NS	NS	NS	NS
Sex °	NS	NS	NS	NS	NS	NS
Preoperative Gissane’s angle ***	*p* < 0.05	*p* < 0.05	*p* < 0.05	NS	NS	*p* < 0.05
Last F.U. Gissane’s angle ***	NS	NS	NS	NS	NS	NS
Preoperative Böhler's angle ***	NS	*p* < 0.05	*p* < 0.05	*p* < 0.05	NS	NS
Last F.U. Böhler's angle ***	NS	NS	NS	NS	NS	NS

* Mann-Whitney U-test, ° Chi-squared test.

## Data Availability

Some or all data and models that support the findings of this study are available from the corresponding author upon reasonable request.

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
