# Peer review of "Patient Reported Outcome Measures (PROMs) in Surgery: Evaluation after Minimally Invasive Reduction and Percutaneous K-Wires Fixation for Intra-Articular Calcaneal Fractures"

_diseases, 2023, doi:10.3390/diseases11020057_

Round 1
Reviewer 1 Report
The article provides the results of a retrospective study with the aim to evaluate Patient Reported Outcome Measures (PROMs) after minimally invasive reduction and percutaneous Kirschner-wires fixation for intra-articular calcaneal fractures (IACF), and to compare PROMs to pre-operative and last follow-up radiographic findings.
The authors found that minimally invasive reduction and percutaneous K-wires fixation provided satisfactory PROMs.
Further studies are needed to confirm their results.
The manuscript is well written and well structured. The text is clear and easy to read. The topic is interesting and in line with the journal.
In my opinion, the article can be published in its current form.
Author Response
The article provides the results of a retrospective study with the aim to evaluate Patient Reported Outcome Measures (PROMs) after minimally invasive reduction and percutaneous Kirschner-wires fixation for intra-articular calcaneal fractures (IACF), and to compare PROMs to pre-operative and last follow-up radiographic findings.
The authors found that minimally invasive reduction and percutaneous K-wires fixation provided satisfactory PROMs.
Further studies are needed to confirm their results.
The manuscript is well written and well structured. The text is clear and easy to read. The topic is interesting and in line with the journal.
In my opinion, the article can be published in its current form.
- The authors would like to thank the reviewer for recognizing the importance of the topic of our research and for kind consideration.
Reviewer 2 Report
Review
Many thanks to the authors for having presented a so interesting study about “Patient Reported Outcome Measures (PROMs) in surgery: evaluation after Minimally Invasive Reduction and Percutaneous K-Wires Fixation for Intra-Articular Calcaneal Fractures”. The language is so good that the manuscript does not need to be corrected by a person of English mother tongue.
Abstract
The abstract is well structured, it contains a summary of the main aims, results and conclusions of the study. Keywords should be sorted in alphabetic order, according to STROBE guidelines.
Introduction
The introduction of the study is well structured, the rationale behind the study is written in a clear and understandable way, moreover it includes the main aims of the study. Also, the comparison between surgical techniques (open reduction vs minimally invasive approaches) gives to the reader a clear overview of the topic. However, a few lines about the difficult treatment of this challenging fractures by different methods over the centuries should be added.
Materials and Methods
This section contains enough information to understand and possibly repeat the study. Almost every aspect of the study has been considered and explained in detail. Inclusion and exclusion criteria were correctly described and listed. Moreover, surgical procedures and postoperative management care was clearly and precisely described, as well as data collection, clinical outcome measures and statistical analysis. One suggestion would be to briefly add some of the evaluation parameters of the PROMs questionnaire administered to patients during follow-up.
Results
The results presented are quite complete, reflecting the M&M section. You could possibly consider adding some explanatory images related to radiographic parameters (e.g., Bohler and Gissane angles) pre-operatively and post-operatively, in order to make it more clear to readers.
Discussion
The length and content of the discussion communicate the main information of the paper, but the results presented have not been discussed adequately with data and references provided in literature. Please, discuss your results with those presented in the most recent literature, quoting.
· Radiographic and functional outcomes after displaced intra-articular calcaneal fractures: a comparative cohort study among the traditional open technique (ORIF) and percutaneous surgical procedures (PS). J Orthop Surg Res. 2016 Aug 22;11(1):92. doi: 10.1186/s13018-016-0426-6.
Also, the authors provided limitations for this study and future potential studies related to the topic necessary to further clarify the matter.
Conclusion
The conclusions reflect and refer to the results of the study. It is written in a schematic way, and it focuses the matters of the study.
The author’s contribution, funding, conflict of interest, informed consent statement, data availability statement was also mentioned in the right way.
References
Most of the references are not up to date. Hence, please delate the one before 2010 if not essential (nr. 7, 8, 11, 12, 17, 18, 20) replacing it with newer one and integrate it with those suggested previously.
Author Response
Many thanks to the authors for having presented a so interesting study about “Patient Reported Outcome Measures (PROMs) in surgery: evaluation after Minimally Invasive Reduction and Percutaneous K-Wires Fixation for Intra-Articular Calcaneal Fractures”. The language is so good that the manuscript does not need to be corrected by a person of English mother tongue.
Abstract
The abstract is well structured, it contains a summary of the main aims, results and conclusions of the study. Keywords should be sorted in alphabetic order, according to STROBE guidelines.
- According to Reviewer’s suggestion we have sorted by alphabetic order the keywords. we thank the reviewer for the useful suggestions
Introduction
The introduction of the study is well structured, the rationale behind the study is written in a clear and understandable way, moreover it includes the main aims of the study. Also, the comparison between surgical techniques (open reduction vs minimally invasive approaches) gives to the reader a clear overview of the topic. However, a few lines about the difficult treatment of this challenging fractures by different methods over the centuries should be added.
- Thanks, we covered the topic suggested from the line 99 to line 129
Materials and Methods
This section contains enough information to understand and possibly repeat the study. Almost every aspect of the study has been considered and explained in detail. Inclusion and exclusion criteria were correctly described and listed. Moreover, surgical procedures and postoperative management care was clearly and precisely described, as well as data collection, clinical outcome measures and statistical analysis. One suggestion would be to briefly add some of the evaluation parameters of the PROMs questionnaire administered to patients during follow-up.
- Thanks for highlighting, we detailed questionnarie and parameters administrated as suggested. L 198 to L 209
Results
The results presented are quite complete, reflecting the M&M section. You could possibly consider adding some explanatory images related to radiographic parameters (e.g., Bohler and Gissane angles) pre-operatively and post-operatively, in order to make it more clear to readers.
- We thank the reviewer for the suggestions; we have added the radiographic images to clarify the text explanation
Discussion
The length and content of the discussion communicate the main information of the paper, but the results presented have not been discussed adequately with data and references provided in literature. Please, discuss your results with those presented in the most recent literature, quoting.
Radiographic and functional outcomes after displaced intra-articular calcaneal fractures: a comparative cohort study among the traditional open technique (ORIF) and percutaneous surgical procedures (PS). J Orthop Surg Res. 2016 Aug 22;11(1):92. doi: 10.1186/s13018-016-0426-6.
Also, the authors provided limitations for this study and future potential studies related to the topic necessary to further clarify the matter.
- We thank the reviewer for the suggestions; we extended the discussion with references provided and add a limitations section
- 297-304 Biz et al. found that patients treated with open reduction and internal fixation techniques presented overall superior radiographic and functional outcomes compared with percutaneous approach. Several minimally invasive techniques were developed for management of intraarticular calcaneal fractures but Ahmed Shams et al. reported that treatment of displaced intraarticular calcaneal fractures with minimally invasive re-duction and fixation using either cannulated screws or K-wires can achieve similar ex-cellent functional and radiological outcomes, with high patient satisfaction and also K-wires has the advantage of reduced operative time.
L.346-356
Limitations and future direction
Studies with larger samples, which include in their methods different surgical procedures are still needed to better investigate the benefits of IACF treated by minimally invasive reduction and percutaneous k-wire fixation and draw more reliable conclusions.
This study is not without limitations: one limitation of the present study concerns the evaluation of effects of surgical procedure on PROMs, the long-term effects has been evaluated at 36 months (mean). For these reasons, further studies should evaluate the effects at one year in order to better understand also the short-term impact of this surgical approach.
Conclusion
The conclusions reflect and refer to the results of the study. It is written in a schematic way, and it focuses the matters of the study.
The author’s contribution, funding, conflict of interest, informed consent statement, data availability statement was also mentioned in the right way.
References
Most of the references are not up to date. Hence, please delate the one before 2010 if not essential (nr. 7, 8, 11, 12, 17, 18, 20) replacing it with newer one and integrate it with those suggested previously.
- references have been updated as suggested by reviewers.
We thank the reviewers for the useful comments and suggestions. The paper has been revised by taking all reviewers’ suggestions, we hope to have improved the manuscript
Reviewer 3 Report
Calcaneus fractures, their treatment and post-surgical functional outcomes are still a problem for orthopedic surgery. The manuscript deals with a much-discussed topic in the available literature which, nevertheless, does not find unanimity.
It is overall well draft and meaningful, however, it needs some revisions to be considered worthy of publication:
-In the introduction, the authors described in detail the disease, emphasizing different surgical approaches and their strengths;
-Materials and methods are complete and the authors make the study reproducible; however, it would be appropriate to move the description of fracture classifications and of the angle to the introduction (lines 127-143);
-Results are clearly presented and the summary tables make understanding fluid and intuitive;
- The discussion should be enriched to emphasize the strengths of the manuscript and differences with similar works (if the authors deem it appropriate, they could add DOI: 10.3390/jcm12010020 and DOI: 10.3390/jcm11195660);
-If possible, the authors should include the clinical-radiographic documentation of some of the cases presented in the study to increase the quality of the article.
Author Response
Calcaneus fractures, their treatment and post-surgical functional outcomes are still a problem for orthopedic surgery. The manuscript deals with a much-discussed topic in the available literature which, nevertheless, does not find unanimity.
It is overall well draft and meaningful, however, it needs some revisions to be considered worthy of publication:
-In the introduction, the authors described in detail the disease, emphasizing different surgical approaches and their strengths;
-Materials and methods are complete and the authors make the study reproducible; however, it would be appropriate to move the description of fracture classifications and of the angle to the introduction (lines 127-143);
- Did it, as suggeted (L.39 to L.59)
-Results are clearly presented and the summary tables make understanding fluid and intuitive;
- The discussion should be enriched to emphasize the strengths of the manuscript and differences with similar works (if the authors deem it appropriate, they could add DOI: 10.3390/jcm12010020 and DOI: 10.3390/jcm11195660);
- We thank the reviewer for the suggestions; we extended the discussion with references provided
- 297-304
Biz et al. found that patients treated with open reduction and internal fixation techniques presented overall superior radiographic and functional outcomes compared with percutaneous approach. Several minimally invasive techniques were developed for management of intraarticular calcaneal fractures but Ahmed Shams et al. reported that treatment of displaced intraarticular calcaneal fractures with minimally invasive re-duction and fixation using either cannulated screws or K-wires can achieve similar ex-cellent functional and radiological outcomes, with high patient satisfaction and also K-wires has the advantage of reduced operative time.
-If possible, the authors should include the clinical-radiographic documentation of some of the cases presented in the study to increase the quality of the article.
- Thanks, we added the radiographic images as you suggested (figure 2)
We thank the reviewers for the useful comments and suggestions. The paper has been revised by taking all reviewers’ suggestions, we hope to have improved the manuscript
Round 2
Reviewer 2 Report
The authors answered my comments properly, improving the manuscript quality. Well done!
Reviewer 3 Report
The authors have taken up our suggestions and made the changes appropriately. The manuscript is, now, worthy of publication.